# Aerosol breezes drive cloud and precipitation increases

Gabrielle R. Leung [1] & Susan C. van den Heever [1]

Aerosol-cloud interactions are a major source of uncertainty in weather and climate models. These interactions and associated precipitation feedbacks are modulated by spatial distributions of aerosols on global and regional scales. Aerosols also vary on mesoscales, including around wildfires, industrial regions, and cities, but the impacts of variability on these scales are understudied. Here, we first present observations of covarying mesoscale aerosol and cloud distributions on the mesoscale. Then, using a high-resolution process model, we show that horizontal aerosol gradients of order 100 km drive a thermally-direct circulation we call an "aerosol breeze". We find that aerosol breezes support initiation of clouds and precipitation over the low-aerosol portion of the gradient while suppressing their development on the high-aerosol end. Aerosol gradients also enhance domain-wide cloudiness and precipitation, compared with homogenous distributions of the same aerosol mass, leading to potential biases in models that do not adequately represent this mesoscale aerosol heterogeneity.

Large-scale global and regional changes in aerosol concentrations have long been understood to be associated with changes in the surface energy budget, cloudiness, and precipitation[1–4]. Aerosol particles influence the amount of energy reaching the surface directly via the extinction and/or absorption of incoming solar radiation[5,6], as well as indirectly via microphysical interactions with clouds[7,8]. These changes subsequently impact surface fluxes, atmospheric warming, and eventually precipitation, with potentially large climatic and societal impacts[1,8].

The spatial distribution of aerosol concentrations is also known to be important. On a global scale, the gradient in the aerosol-radiative effect between the northern and southern hemispheres influences the location of the ITCZ and its associated precipitation maximum[9,10]. On a regional scale, changes to monsoon circulations and precipitation have similarly been attributed to gradients in aerosol emissions and their direct effects in areas such as East, South, and Southeast Asia[11–15]. On smaller scales, variability in horizontal aerosol concentrations on the order of 40–400 km has frequently been observed in concert with major aerosol sources, sinks, and transport pathways[16–18]. For example, horizontal aerosol gradients typically exist on the edges of wildfire smoke plumes[19,20] or urban areas[21,22]. However, little work has been done

linking those mesoscale aerosol gradients to cloud or precipitation processes.

Mesoscale variations in surface fluxes due to contrasts in surface properties (e.g., sea breeze, forest breeze, slope flows) are known to drive thermal circulations that are important for organizing and enhancing cloudiness and precipitation[23–25]. Furthermore, in the tropics where synoptic pressure gradients are generally weak, mesoscale pressure gradients due to differential heating play a strong role in defining wind patterns and convection[26–28]. Horizontally-uniform changes to aerosol loading are also known to impact the strength of mesoscale circulations by influencing incoming solar radiation[29,30]. It is thus conceivable that mesoscale horizontal gradients in aerosol concentration over an otherwise uniform land or ocean surface may drive thermal circulations similar to sea breezes, thereby enhancing cloudiness and precipitation. We will refer to such aerosol gradient-induced circulations as "aerosol breezes".

There is limited observational[31] and modeling[32] evidence for the existence of an "aerosol breeze" due to light-absorbing aerosol. Lee et al. (2014) confirmed that spatial gradients in absorbing aerosol concentrations could generate circulation patterns influencing cloud formation, both by reducing the amount of radiation reaching the surface and by changing the static stability of the boundary layer.

[1]Department of Atmospheric Science, Colorado State University, Fort Collins, CO, USA. ✉e-mail: gabrielle.leung@colostate.edu

These two effects act in opposite directions, with the net impact on the circulation depending on the location and magnitude of the gradient. However, we expect aerosol particles which are primarily scattering (e.g., sulfates) may similarly increase extinction, but would produce relatively minor changes to the static stability of the atmosphere. Thus, the impacts of spatial gradients in scattering aerosol concentration on cloud systems may be even more pronounced.

In this paper, we study the impacts of mesoscale horizontal variability in sulfate aerosol concentrations on the frequency, distribution and precipitation amounts of shallow convective clouds. Our goal is to determine whether mesoscale gradients in sulfate aerosol can drive aerosol breezes, and if so, what the subsequent impacts of aerosol breezes are on clouds and precipitation. We also aim to assess the implications and potential biases introduced by failing to resolve these mesoscale aerosol gradients in larger-scale regional and climate models. Given that aerosol breezes are likely to become increasingly important with changing climates as the wildfire risk continues to increase[33] and emissions from urban and industrial regions shift in their spatial patterns[34], it is critical that we understand and appropriately forecast such effects.

## Results
### Observational case studies

We first present two examples of aerosol loadings and cloud cover covarying over distances on the order of 10–100 km (mesoscale) in Fig. 1. While the two cases differ in terms of geographic location, land

surface type, and prevailing synoptic meteorology, they are remarkably similar in terms of the observed patterns on cloudiness relative to the location of the aerosol gradient. Thus, we present them here as possible examples of the aerosol breeze phenomenon.

The first case (Fig. 1a, b) involves a likely aerosol breeze over Kentucky and central Tennessee, USA on 3 July, 2021 associated with wildfire smoke advecting into the region. The wildfire smoke formed a strong aerosol gradient between the regions labeled "high-aerosol" and "low-aerosol". AODs ranged between <0.01 in the low-aerosol region to ~0.5 in the high-aerosol region over an area of a few hundred kilometers. Throughout the day, shallow cumulus clouds developed in the low-aerosol region.

The second case (Fig. 1c) shows an image captured from the International Space Station over southwestern Australia on 12 January 2020. A smoke plume from an active fire is located in the center of the image, with smoke being advected to the east/southeast over a distance on the order of a 100 km. Along the northeast corner of the image, a field of shallow clouds developed only in the clear-air or low-aerosol region along the edges of the smoke plume.

These case studies motivate an investigation into the underlying physics driving such cases. Apart from the gradient in aerosol emissions, there are no clear synoptic forcing or surface heterogeneities aligned with the spatial distribution and extent of the shallow cumulus fields in the two cases presented. Thus, we suspect that the juxtaposition of gradients in aerosol loading and cloudiness point to the potential influence of a mesoscale circulation like an aerosol breeze,

## Case 1: Central USA, 2021-07-03 10:30 LT

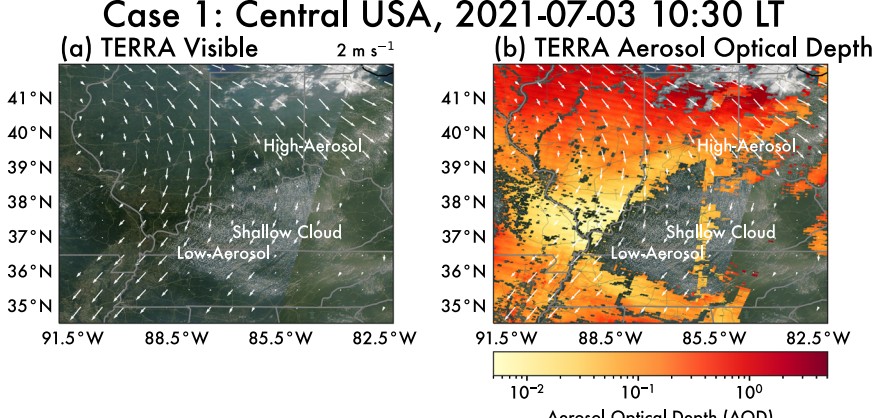

## Case 2: Southwestern Australia, 2020-01-12 17:22 LT

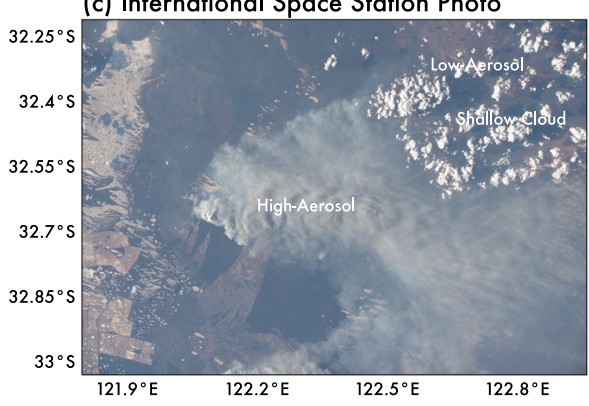

**Fig. 1 | Observational cases of covarying aerosol and cloud fields.** Two observational examples of a strong aerosol gradient with associated shallow cumulus formation. The top row (**a, b**) shows smoke advection over Kentucky and central Tennessee, USA on 3 July 2021. **a** Visible channel and **b** aerosol optical depth (AOD; 3 km resolution) from Terra MODIS. Note that regions in (**b**) with no data are where no AOD was retrieved due to cloud cover. 2-m wind vectors in (**a, b**) are taken from MERRA-2 for the same time period as the satellite imagery. The bottom row (**c**) shows smoke advection over southwestern Australia on 12 January 2020. The image was captured from the International Space Station (image ID: ISS061-E-123446, accessed from eol.jsc.nasa.gov courtesy of the Earth Science and Remote Sensing Unit, NASA Johnson Space Center). Regions of high- and low-aerosol loading, as well as shallow cloud cover are annotated.

but further analysis of the fundamental processes involved in such a circulation are needed.

## Aerosol breeze circulation

To investigate the dynamical processes driving such mesoscale aerosol gradients and associated patterns of cloudiness, we set up a suite of simple idealized large eddy simulations (LES) using the Regional Atmospheric Modeling System (RAMS)[35,36]. The model domain covered an area equivalent to the size of a typical global climate model (GCM) grid cell. The *Gradient* simulation was initialized to be horizontally homogenous in terms of land surface and meteorology, but had a meridional gradient in surface light-scattering aerosol emissions between 100 and 1000 cm$^{-3}$ day$^{-1}$ (see "Methods").

The two observational cases presented earlier are certainly more complex than this idealized modeling scenario. Aerosol plumes closer to a point source (as in Fig. 1c) tend to be more conical and are impacted by dilution moving away from the source; this would lead to aerosol gradients both parallel and perpendicular to the direction of plume advection. However, as our goal is to conceptually understand the fundamental physical processes at play, we limited the complexity of the model set-up in order to isolate the effects of an aerosol gradient on the mesoscale circulation to a single dimension. Thus, the *Gradient* simulation is more similar to smoke that has been transported over a long range as in Fig. 1a, b, and represents aerosol gradients that might be observed at some distance from a smoke plume, an urban region, or other mesoscale aerosol sources. After first identifying the fundamental processes governing how mesoscale aerosol gradients can influence cloud and precipitation formation in this idealized model set-up, the processes elucidated here can then be generalized to more complex distributions of aerosol loading.

Within 4 h from initialization, a distinct circulation forms between the low- and high-aerosol regions of the domain (Fig. 2). Note that

while the high-aerosol region is in the center of in the actual model domain (see "Methods"), the cross-section in Fig. 2 is constructed such that the abscissa is given in terms of distance from the center line where aerosol concentrations are highest (i.e., the domain has been "folded" to more clearly demonstrate the circulation, described in "Methods"). Aerosol optical depth is highest in the center of the domain (left side of Fig. 2c), in line with the prescribed aerosol gradient. As a result of the increased light extinction, the downwelling shortwave at the surface is ~30 W m$^2$ lower in the high-aerosol region compared to the low-aerosol region (Fig. 2b). This causes uneven heating of the land surface, with greater surface temperatures occurring in the low-aerosol region (Fig. 2c). A gradient in surface fluxes forms opposite to the direction of the gradient in aerosol (Fig. 2d). This drives a pressure gradient and net wind flow directed from the high- to low-aerosol region at altitudes between the surface and the top of the surface-based mixed layer/cloud base (Fig. 2a, b). Above those altitudes, there is a compensating return flow aloft up to ~2 km. Although there is still a difference in the downwelling shortwave flux above those altitudes (Fig. 2b), the height of the return flow is limited by the increase in static stability at the tropical trade wind inversion (~2 km AGL) (see "Methods"). The resultant low-level convergence favors rising motion over the low-aerosol region, and subsiding motion over the high-aerosol region (Fig. 2a). The aerosol-induced circulation that develops is thus an aerosol breeze, and is similar to other thermally-driven mesoscale circulations such as sea breezes, both in terms of its thermal driving mechanism and general structure[25,37].

## Precipitation response

The aerosol-induced circulation just described leads to the preferential development of clouds and precipitation over the low-aerosol region of the aerosol gradient (Fig. 3). Clouds forming over the low-aerosol region have greater coverage, higher cloud tops (Fig. 3b), and are more

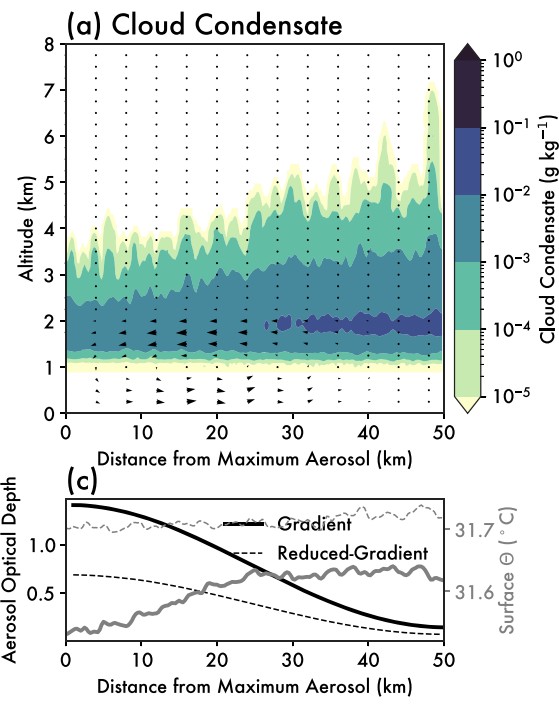
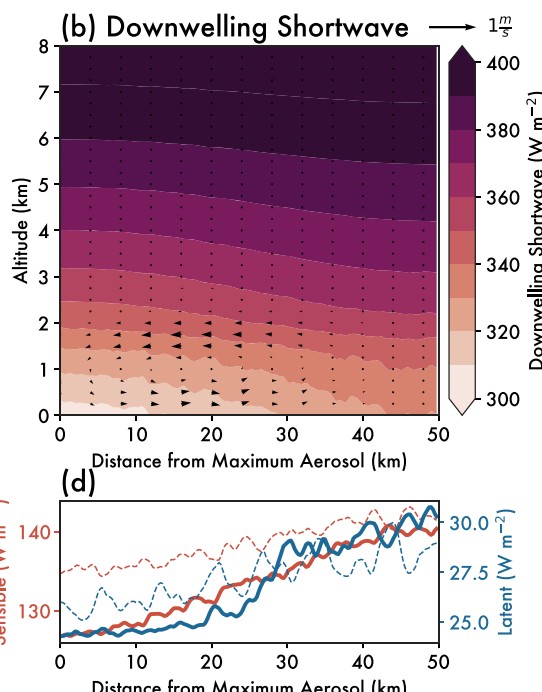

**Fig. 2 | Aerosol breeze circulation from scattering aerosol gradient simulation.** Mean cross section through the domain over all 12 h of the scattering aerosol gradient simulation, averaged temporally and zonally. The abscissa is given as a function of distance from the domain center/maximum aerosol concentration (horizontal black line in Fig. 6b, c), such that the high-aerosol region is on the left and the low-aerosol region is on the right of these panels. Shading in (**a**) shows cloud condensate mixing ratios (g kg$^{-1}$), and in (**b**) the downwelling shortwave flux (W m$^{-2}$). The wind barbs in (**a**) and (**b**) show the mean vertical and horizontal

winds oriented along the aerosol gradient. Vertical winds are multiplied by a factor of 5 so as to be more visible in the figure. **c** Depicts the aerosol optical depth at the surface in black (left y-axis) and the surface potential temperature (θ) in gray (right y-axis; °C), while **d** depicts the sensible surface heat flux in red (left y-axis; W m$^{-2}$) and latent surface heat flux in blue (right y-axis; W m$^{-2}$), all averaged temporally and zonally. The dashed lines in (**c**, **d**) show the same quantities for the *Reduced-Gradient* simulation (see "Methods"). Source data are provided as a Source data file.

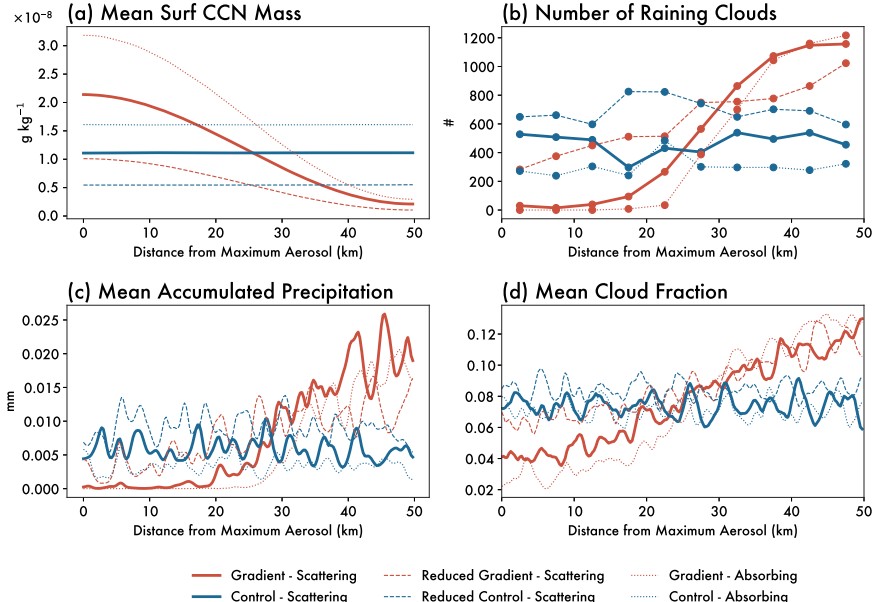

**Fig. 3 | Domain-wide differences between *Gradient* and *Control* simulations.** Comparison of *Gradient* (red) and *Control* (blue) simulations in terms of **a** number of clouds with mean precipitation rates of at least 0.1 mm h⁻¹, **b** mean accumulated precipitation per grid cell (mm), **c** mean surface aerosol mass concentration (g kg⁻¹), and **d** mean cloud fraction. The quantities are shown in solid lines for scattering aerosol, dashed lines for reduced aerosol, and dotted lines for absorbing aerosol simulations. Source data are provided as a Source data file.

likely to produce rain (Fig. 3a) compared to clouds over the high-aerosol region of the gradient. Almost none of the rain throughout the entirety of the simulation falls over the high-aerosol region (Fig. 3b), instead being concentrated over the low-aerosol region.

To assess the influence of the spatial distribution of aerosol particles, we simulated a *Control* case with the same integrated aerosol mass and number concentrations as in the *Gradient* simulation but now distributed uniformly in the horizontal. In comparing the *Gradient* and *Control* simulations, we are able to isolate the impacts of the aerosol gradient separately from the impacts of the aerosol loading itself.

Relative to the *Control*, the additional low-level convergence driven by the aerosol gradient increases the number of precipitating clouds (Fig. 4a) and the accumulated precipitation (Fig. 4b) produced within the domain. The onset of rain also occurs an hour sooner in the presence of a strong aerosol gradient relative to *Control* (Fig. 4), and the difference in domain-wide accumulated precipitation between the two simulations is ~25% after 12 h (Fig. 4b). This demonstrates that the aerosol breeze is further analogous to other mesoscale flows driven by surface heterogeneities in its capacity not only to redistribute convection throughout the domain, but also to actually increase it. Studies examining the interactions between aerosols and clouds often do not realistically represent the spatial heterogeneity in aerosol gradients. Thus, their estimated aerosol-cloud-precipitation interactions may not accurately reflect the magnitude of such effects in regions of strong aerosol gradients. Furthermore, this result suggests that unrepresented aerosol heterogeneities across climate and regional models with grid spacings that are coarse relative to the scale of the aerosol gradient may lead to biases in rain timing, distribution, and even total rain amount. Limited area models at higher resolutions may be able to capture the effect of the mesoscale secondary circulations described here, but only if the spatial gradients of aerosol concentration are properly represented.

**Sensitivity to aerosol loading**
The maximum AOD in the *Gradient* simulation is on the high end of observed aerosol loadings, as might be expected close to a large major aerosol source like an intense wildfire. We additionally tested the

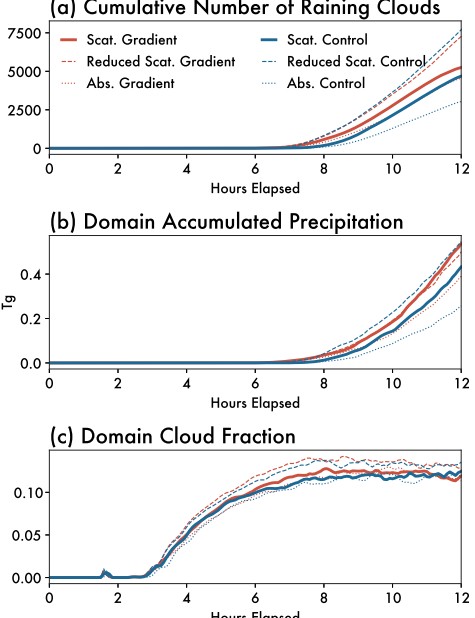

**Fig. 4 | Temporal differences between *Gradient* and *Control* simulations.** Timeseries comparisons of the *Gradient* (red) and *Control* (blue) simulations in terms of **a** cumulative number of raining clouds (mean rain rate of at least 0.1 mm h⁻¹), **b** total domain accumulated precipitation (Tg), and **c** domain cloud fraction over the 12 h of the simulation. Scattering aerosol simulations are indicated using solid lines, reduced aerosol simulations using dashed lines and absorbing aerosol simulations using dotted lines. Source data are provided as a Source data file.

sensitivity of the idealized aerosol breeze circulation to the magnitude of the aerosol loading by conducting a set of *Reduced* simulations with half the aerosol loading as in the initial simulations (i.e., the surface aerosol emissions ranged from 50 to 500 cm⁻³ day⁻¹). A qualitatively

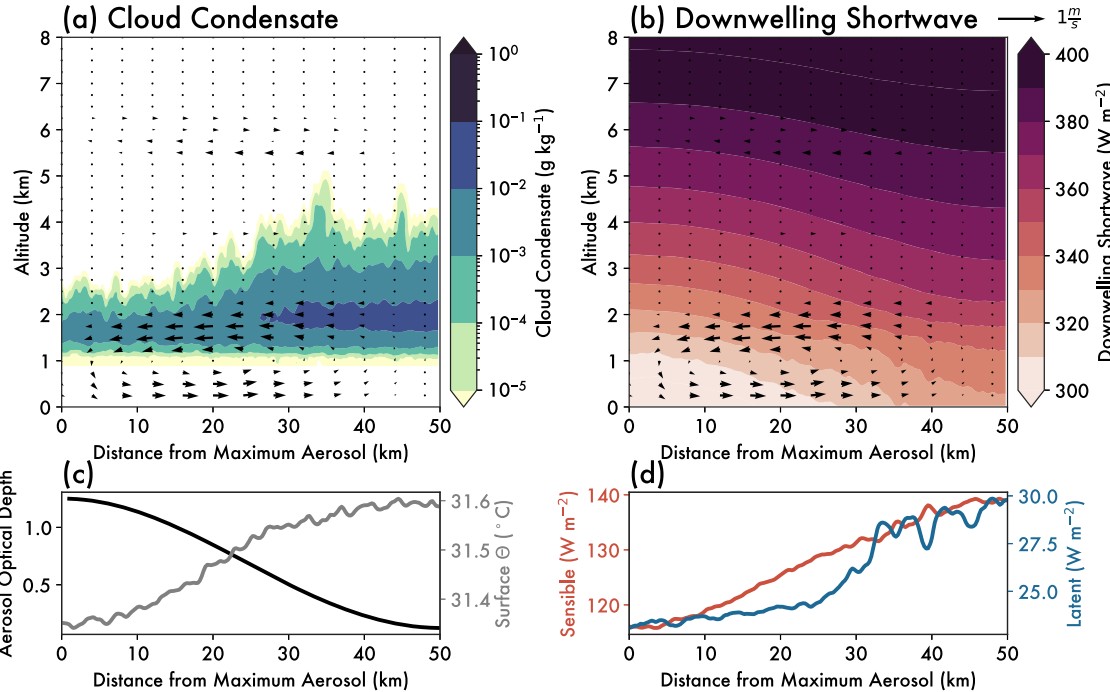

**Fig. 5 | Aerosol breeze circulation from absorbing aerosol gradient simulation.** Mean cross section through the domain over all 12 h of the absorbing aerosol gradient simulation, averaged temporally and zonally. The abscissa is given as a function of distance from the domain center/maximum aerosol concentration (horizontal black line in Fig. 6b, c), such that the high-aerosol region is on the left and the low-aerosol region is on the right of these panels. Shading in (**a**) shows cloud condensate mixing ratios (g kg⁻¹), and in (**b**) the downwelling shortwave flux (W m⁻²). The wind barbs in (**a**) and (**b**) show the mean vertical and horizontal winds oriented along the aerosol gradient. Vertical winds are multiplied by a factor of 5 so as to be more visible in the figure. **c** Depicts the aerosol optical depth at the surface in black (left y-axis) and the surface potential temperature (θ) in gray (right y-axis; °C), while **d** depicts the sensible surface heat flux in red (left y-axis; W m⁻²) and latent surface heat flux in blue (right y-axis; W m⁻²), all averaged temporally and zonally. The dashed lines in (**c**, **d**) show the same quantities for the *Reduced-Gradient* simulation (see "Methods"). Source data are provided as a Source data file.

similar circulation develops under the *Reduced-Gradient* simulations (Supplementary Fig. 1), though the magnitude of the gradient in surface fluxes is reduced in proportion to the reductions in the AOD contrast between low- and high-aerosol portions of the domain (Fig. 2c, d).

The increase in convection and precipitation associated with the aerosol breeze is sensitive to the magnitude of the aerosol gradient. Although an aerosol breeze does develop and subsequently increases clouds and precipitation in the low-aerosol region (Fig. 2a–c) of the *Reduced-Gradient* simulation, a smaller change in the total amount of precipitation is produced relative to the *Reduced-Control* simulation (Fig. 3a, b). This suggests the net impact of the aerosol gradient on accumulated precipitation results from the competition between direct[3,5,6] (i.e., reduction in radiation over high-aerosol regions) and indirect[7,8] (i.e., microphysical invigoration of warm clouds over high-aerosol regions) effects. As such, the *Gradient* simulation represents an estimated upper bound on the magnitude of an aerosol breeze and associated cloud and precipitation formation, where the direct effect far outweighs contributions from the indirect effect, and the aerosol breeze circulation that develops is clearly pronounced.

## Sensitivity to aerosol type

Because the aerosol breeze is primarily driven by direct aerosol effects (i.e., aerosol-radiation interactions), it stands to reason that the aerosol breeze may be sensitive to the radiative properties of the aerosol being emitted. To test this, we conducted an additional set of simulations with light-absorbing (absorbing carbon) rather than light-scattering (ammonium sulfate; shown in Fig. 2) aerosol. The major difference in radiative properties between these two aerosol types is that scattering aerosol predominantly impacts the extinction of downwelling radiation (by scattering light away from the surface), whereas absorbing aerosol have an additional atmospheric impact due to absorbing and

reemitting longwave radiation[11,32]. The remitted longwave radiation leads to changes in static stability by locally warming the atmosphere in regions where the aerosol is abundant.

We find that a gradient in absorbing aerosol (Fig. 5) drives a similar aerosol breeze as a gradient in scattering aerosol does. There is also a similar increase in cloudiness and precipitation (dotted lines in Fig. 3) on the low-aerosol side of the domain, and up to a ~50% increase in domain-wide precipitation after 12 h relative to a control simulation (dotted lines in Fig. 4). Due to warming and resultant increases in static stability aloft, cloud development in the absorbing aerosol simulation is capped at approximately ~5 km (Fig. 5a). We would thus expect that the specific properties of the aerosol breeze depend on the height of the aerosol loadings, particularly for absorbing aerosol. In the case presented here, aerosol is emitted from the surface, but lofted aerosol layers would lead to a different vertical structure of static stability[38] that might interact differently with the aerosol breeze. Nonetheless, the development of an aerosol breeze resulting from gradients in both scattering and absorbing aerosol emissions demonstrates that the aerosol breeze concept is relatively robust, though the specific characteristics of the aerosol breeze will likely depend on the dominance of scattering or absorbing aerosol in the region of interest.

In conclusion, these results demonstrate that a sufficiently strong gradient in aerosol concentrations alone can drive an "aerosol breeze" that impacts mesoscale circulations, cloud properties, and precipitation, even in the absence of other mesoscale heterogeneities in surface properties. Furthermore, this holds true for both light-scattering and light-absorbing aerosol. Failing to represent the radiative impacts and resulting circulations induced by mesoscale (and hence sub-grid scale) horizontal aerosol gradients within GCM and regional models may lead to significant biases in the predicted timing and amounts of cloudiness and precipitation. It may also lead to biases in the estimation of

aerosol-cloud-precipitation feedbacks in other higher-resolution studies which do not appropriately represent the spatial heterogeneity of aerosol emissions, either because of sub-grid scale gradients or because the aerosol concentration is otherwise assumed to be homogenous across the domain. The two observational cases presented here associated with the advection of thick wildfire smoke are similar in spatial scale, aerosol gradient set-up, and cloud formation to our idealized modeling case, and further support the fact that the impact of localized aerosol emissions on mesoscale phenomena as a result of the direct aerosol effect is important and should be considered.

The modeling results presented here serve as an upper bounding case to demonstrate the primary physical processes involved in producing the aerosol breeze, in a relatively idealized set-up with a strong and well-defined gradient in aerosol isolated from other heterogeneities in land surface or meteorology. This does not imply that other factors such as synoptic forcing may not also play a role in such circulations, and future work should focus on case study modeling in an attempt to assess such roles. Our findings emphasize the need for greater consideration of aerosol breezes in future work, particularly in investigating the sensitivity of aerosol breezes to meteorology, aerosol type, land surface, varied spatial distributions (both horizontal and vertical), and interactions with other circulations (such as sea breezes or buoyant firestorm plumes). Such research is particularly essential given projected changes globally to the spatial distribution of aerosol emissions in urban and industrial areas, as well as in wildfires, with changing climates.

## Methods

We used the Regional Atmospheric Modeling System (RAMS, version 6.3.02) to run the simulations in this study[35,36]. RAMS is a non-hydrostatic atmospheric model with a sophisticated two-moment bin-emulating bulk microphysics scheme; full representation of aerosol sources, sinks, and advection; coupled surface fluxes using the LEAF-3 submodel; and an interactive two-stream radiation scheme including aerosol-radiative effects. Further details about the model set-up can be found in Table 1 and associated references.

Our model grid spanned $100 \times 100$ km horizontally—similar in size to a single $1 \times 1°$ GCM grid box—at a horizontal spatial resolution of 100 m. In the vertical direction, the model grid was 20 km tall, with spacing stretching between 50 and 300 m to resolve the cumulus

cloud field and boundary layer processes. The simulation is idealized, but broadly intended to represent summer monsoon conditions in the Maritime Continent. Initial conditions were based on the mean ERA-5 profile over a $2 \times 2°$ box over Luzon Island of the Philippines during September 2019 to coincide with the Cloud, Aerosol, and Monsoon Process Philippines Experiment (CAMP$^2$Ex) (Fig. 6a, b). We also

**Table 1 | Regional Atmospheric Modeling System (RAMS) model options used in simulation**

| Model Aspect | Setting |
|---|---|
| Grid | Arakawa C grid |
| | 1000 × 1000 points, $\Delta x = \Delta y = 100$ m |
| | 120 vertical levels, $\Delta z = 50$–300 m |
| Time integration | 12 h simulation duration, $\Delta t = 1$ s |
| | Output analysis files every 5 min |
| Initialization | Horizontally homogenous thermodynamic profile, averaged from ERA-5 as described in text |
| | No initial background winds |
| | Random potential temperature perturbations within the lowest 500 m AGL of the domain, with a maximum perturbation of 0.1 K |
| Surface scheme | Uniform surface of evergreen broadleaf tree and silty clay loam soil |
| | LEAF-3[39] |
| Boundary conditions | Periodic in zonal and meridional directions |
| Microphysics scheme | Two-moment bulk microphysics[40] |
| | 8 hydrometeor classes[41] |
| Radiation scheme | Two-stream, hydrometeor sensitive[42] |
| | Updated every 1 min |
| Aerosol treatment | Ammonium sulfate and absorbing carbon aerosol, with single log-normal mode |
| | Varying concentration in the horizontal as depicted in Fig. 6c |
| | Maximum concentration at the surface and exponentially decreasing with altitude |
| | Aerosol-radiation interactions on |
| | Aerosol sources and sinks on, with full aerosol budget tracking |

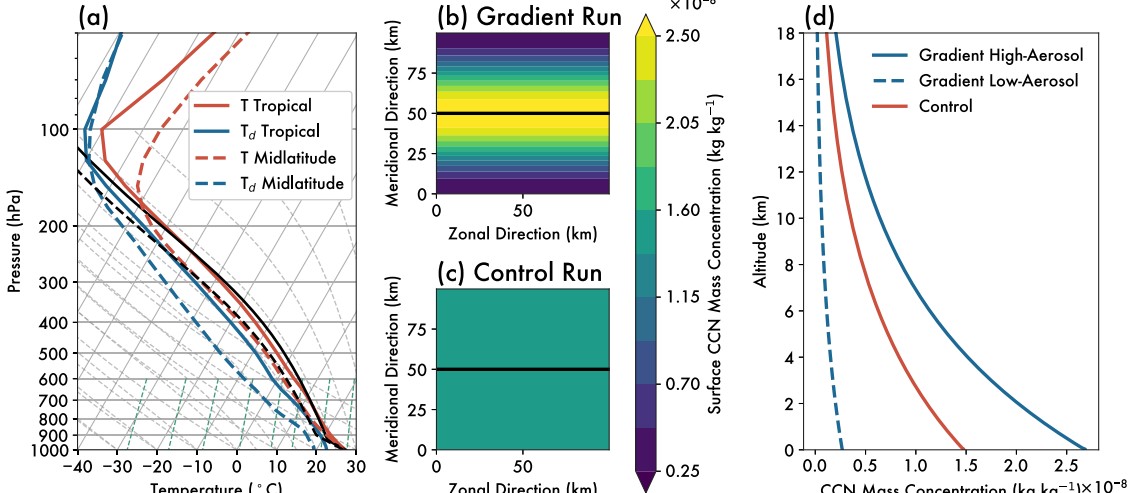

**Fig. 6 | Model set-up details. a** Skew$T$ − log$p$ diagram showing the sounding used to initialize the numerical simulation. The black line is a parcel trajectory from the surface. Source data are provided as a Source data file. Plan view of the surface aerosol mass concentration are shown for the **b** *Gradient* run and **c** *Control* run. The black horizontal line in (**b**) and (**c**) indicates the center of the domain and aerosol gradient (peak aerosol loading), as described in the text. **d** Vertical profile of aerosol mass concentration for the control run (red line), and high- (blue solid line) and low-aerosol (blue dashed line) regions of the gradient run.

conducted identical simulations using a mean sounding over the central USA during July 2021 (corresponding to Case Study 1 in Fig. 1a, b), with results shown in Supplementary Figs. 2–3. The impact of the aerosol gradient in the midlatitude simulations is similar to that in the tropical simulations, with an aerosol breeze developing within a few hours. We have thus opted to present only the tropical simulations in the main text for simplicity, though the impact of different meteorological conditions on the properties of the aerosol breeze is certainly a valuable area of future study.

The initial horizontal aerosol gradient followed a sine curve meridionally, such that aerosol concentration was maximized in the center of the domain and fell off smoothly towards the domain edges (Fig. 6b). The cross-sections presented in Figs. 2 and 5 are taken with respect to this center line (black line in Fig. 6b). The initial aerosol number concentration at the surface ranged from 100 to 1000 cm$^{-3}$. Aerosol concentrations were zonally uniform and decreased exponentially in the vertical (Fig. 6d). This run is referred to as the *Gradient* run. For comparison, we also ran a *Control* simulation with an integrated aerosol mass and number equal to the *Gradient* run but distributed homogenously in the horizontal (Fig. 6c). In both simulations, the aerosol gradient was maintained via a source function identical to the initial aerosol concentrations in the respective simulations over the first 1 km AGL, with the initial aerosol concentration replenished on the timescale of a day. To test the sensitivity of the results to the magnitude of aerosol concentrations, we performed an additional set (*Gradient* and *Control* runs) of simulations with reduced aerosol concentrations. In these reduced-concentration runs, the initial aerosol number concentration at the surface ranged from 50 to 500 cm$^{-3}$. We primarily present results from the initial set of simulations, but where relevant, we refer to these set of sensitivity tests as *Reduced* runs (i.e., *Reduced-Gradient* and *Reduced-Control* as opposed to the *Gradient* and *Control* runs). Both light-scattering (ammonium sulfate) and absorbing (absorbing carbon) aerosol are tested.

After initialization, the simulation was allowed to evolve without additional large-scale forcing except the aerosol emissions. The diurnal cycle was not represented, and the sun was kept at a constant solar zenith equivalent to local solar noon to facilitate the analysis. However, we also conducted a simulation with the diurnal cycle represented and found that a qualitatively similar aerosol breeze circulation developed within 2 h of sunrise (Supplementary Fig. 4).

To count and compare the number of updrafts in different regions of the domain, we used the *tobac* (Tracking and Object-Based Analysis of Clouds, version 1.4) algorithm, which can identify and track updrafts through time[43,44]. The updraft features are first identified in three-dimensions as relative maxima in vertical velocities above multiple threshold values (1, 3, 5 m s$^{-1}$). These updrafts are then linked in time by matching features in previous timesteps based on the predicted updraft motion. We excluded any features that had a lifetime of less than 5 min (i.e., the cloud feature had to be identified in at least two consecutive output files).

## Data availability

The model data used in this study are available from the corresponding author on request, due to the large file sizes involved. The simulations described here can also be reproduced using the information and source code described in the "Code availability" statement below. The satellite data from Terra MODIS are available at https://doi.org/10.5067/MODIS/MOD04_L2.006[45]. The reanalysis data from MERRA-2 are available at https://doi.org/10.5067/VJAFPLI1CSIV[46]. Source data are provided with this paper.

## Code availability

RAMS model, analysis, and plotting code are available on Github at https://doi.org/10.5281/zenodo.7562992[47].

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

## Acknowledgements
This research was supported by NASA CAMP²Ex Grant 80NSSC18K0149 (S.C.v.d.H., G.R.L.).

## Author contributions
G.R.L. and S.C.v.d.H. designed and conceptualized the experiments. G.R.L. conducted the RAMS simulations and wrote the analysis code. G.R.L. and S.C.v.d.H. performed the data analysis and prepared the paper.

## Competing interests
The authors declare no competing interests.
