## [Peer Review File · Nature Communications]

Aerosol breezes drive cloud and precipitation increasesREVIEWER COMMENTS

Reviewer #1 (Remarks to the Author):

Key results:

This study proposes a new concept called "aerosol breeze". It occurs when a gradient in aerosol concentration exists on the order of 100 km. Such a gradient produces differential heating to the surface, and thereby results in a local circulation similar to sea breeze. Compared to those in the region with high aerosol concentration, the clouds and precipitations are enhanced in the region with low aerosol concentration.

Originality and significance:

As far as I know, this is the first study proposing the "aerosol breeze" concept. However, the significance of aerosol breeze is not clear, based on the present study. Please see the comments below.

Validity:

I have some major concerns on the validity of this study.

First, the observational case studies lack in-depth analysis. The authors argue that the phenomena shown in Figure 1 are due to aerosol breeze. However, as pointed out by the authors from 97-102, there are other factors such as synoptic forcing and surface heterogeneities (in vegetation, topography, etc.) that can produce similar results. These factors should be carefully analyzed and included as part of the paper, instead of simply stating that these factors are not important.

Second, there are logical inconsistencies in the experimental design. The two case studies are from the Australia and the USA, both of which are in the mid-latitudes and over the continent. While the sounding used to initialize the simulation is from Maritime Continent, which is in the tropical and is characterized by many islands. Over the Maritime Continent, the effect of aerosol breeze is probably masked by the effect of sea breeze. What further confuses me is that the potentially important effect of land-sea contrast is not considered in the simulations.

Third, the treatment of solar radiation and aerosol concentration is questionable. The authors set the solar radiation to be constant at the solar noon. This might unrealistically exaggerate the effect of aerosol breeze. Furthermore, as shown in Figure 4, the effect of aerosol breeze is sensitive to the aerosol concentration. It seems reasonable to anticipate that the effect of aerosol breeze is also sensitive to the solar radiation. In addition, the aerosol concentration of the "reduced" simulations is not clear.

Fourth, the two observational studies focus on smoke aerosols, which usually have a high percentage of light-absorbing aerosols. However, the aerosol in the simulation is assumed to be scattering, only.

References:

Sokolowsky et al., 2022, which is cited at line 276, is not found in the references.

Reviewer #2 (Remarks to the Author):

Review on the "Aerosol breezes drive cloud and precipitation increases" by Leung and van den Heever

This paper argues that effect of the horizontal gradient of aerosols which is known as the aerosol breeze effect has long been underappreciated, however it can be a significant factor to better understand the aerosol-cloud interaction. The paper first shows two satellite observed cases to build the case, then it conducts a numerical study with a high-resolution model to show the aerosol breeze effect. The paper is well organized and well stated. The methodology is clear and is scientifically sounding. Only major issue I found is the role of the absorbing aerosols, which is overlooked in the consideration throughout the manuscript. The study only considers scattering aerosols for the aerosol breeze effect. The two observation cases are indeed the wildfires, which naturally contain quite significant amount of light absorbing aerosols, such as black carbon and organic carbon aerosols. The paper should clearly state 1) how significant the role of the light absorbing aerosol in the aerosol breeze effect, 2) why wildfire cases are qualifying to demonstrate

the role of the aerosol breeze cases. I would imagine that urban conditions are dominated by scattering aerosol and the aerosol breeze over those regions can be detected from satellite observations. I would like to suggest major revision for the Nature Communications upon the response to my major comment.

Minor comments

Line 52: Lee et al. (2014), citation is missing.

Line 80, 86-87 (Figure 1a,b,c): "low-aerosol" It needs to be more elaborated. Low-aerosol region is also cloud covered; therefore it is more-like missing AOD. I believe MODIS cannot observe AOD in cloudy conditions. How one can tell AOD under cloudy conditions? Also I would like to suggest including wind vector to each case 1) to understand background conditions, and 2) to make sure if they (i.e., aerosol and cloud fields) are one dynamic system or two different systems.

Figure 2: Please show the unit of the wind barbs. Vertical wind looks quite weak.

Figure 3-4: How significant the overall impact of aerosol breeze, e.g., compared to the impact of absorbing aerosols? Cloud fraction seems a useful variable to examine the aerosol breeze, however it is not considered in the result. I would suggest looking into cloud fraction in the revision.

Line 219-223: Again, light absorbing aerosols are missing in the discussion.

Reviewer #1 (Remarks to the Author):

Key results:

This study proposes a new concept called “aerosol breeze”. It occurs when a gradient in aerosol concentration exists on the order of 100 km. Such a gradient produces differential heating to the surface, and thereby results in a local circulation similar to sea breeze. Compared to those in the region with high aerosol concentration, the clouds and precipitations are enhanced in the region with low aerosol concentration.

Originality and significance:

As far as I know, this is the first study proposing the “aerosol breeze” concept. However, the significance of aerosol breeze is not clear, based on the present study. Please see the comments below.

We thank the reviewer for their insightful comments and suggestions, which have greatly helped us to improve the quality of this manuscript. We provide responses below in blue font.

Validity:

I have some major concerns on the validity of this study.

First, the observational case studies lack in-depth analysis. The authors argue that the phenomena shown in Figure 1 are due to aerosol breeze. However, as pointed out by the authors from 97-102, there are other factors such as synoptic forcing and surface heterogeneities (in vegetation, topography, etc.) that can produce similar results. These factors should be carefully analyzed and included as part of the paper, instead of simply stating that these factors are not important.

The goal of this paper was to examine whether gradients in aerosol can drive local circulations, and if so, to analyze the aerosol-radiation processes that govern such circulations. To do this, we started by providing two examples in which we suspect that aerosol breezes are important to the cloud features observed. We did this in order to demonstrate that such flows may exist in reality. Then, in order to understand the possible role played by aerosols in driving these physical circulations at the most basic level, we opted to run idealized model simulations that allow us to focus exclusively on the process of interest. Because these idealized simulations are not limited to a single case study and rather represent more generalized physical processes and situations, our findings and conclusions are more generic in application.

The reviewer suggests that other factors (e.g. vegetation, topography) may be able to produce similar physical circulations. We completely agree with the reviewer that other factors may well be important, and did, as the reviewer notes, point this out in the original manuscript. More detailed case study analyses, as suggested by the reviewer, should indeed form an important next step in this research. However, these next steps cannot be undertaken without first understanding whether it is indeed possible for aerosol gradients to drive such circulations, and if so, to formulate a working hypothesis as to how such circulations physically work—these two goals comprise the overarching focus of this paper. This being said, we have highlighted these points in the conclusion in lines 263-269: “This does not imply that other factors such as synoptic forcing may also play a role in such circulations, and future work should focus on case study modeling in an attempt to assess such roles. Our findings emphasize the need for greater

consideration of aerosol breezes in future work, particularly in investigating the sensitivity of aerosol breezes to meteorology, aerosol type, land surface, varied spatial distributions (both horizontal and vertical), and interactions with other circulations (such as sea breezes or buoyant firestorm plumes).”

Second, there are logical inconsistencies in the experimental design. The two case studies are from the Australia and the USA, both of which are in the mid-latitudes and over the continent. While the sounding used to initialize the simulation is from Maritime Continent, which is in the tropical and is characterized by many islands.

Again, we want to state that the two examples presented are just that, and were not intended to be used as case study simulations. However, the reviewer does make a good point that differences in meteorological conditions between the tropics and midlatitudes may influence characteristics of the aerosol breeze we describe in this paper. The model simulation based on a sounding from the Maritime Continent is merely intended as one instance of the aerosol breeze concept. However, in order to address the reviewer’s concerns and demonstrate the generality of the aerosol breeze phenomenon across different meteorology, we have conducted an additional set of simulations using a midlatitude sounding. For the reviewers’ reference, the following figure (included in the Supplementary Information) is the equivalent of Figure 2 in the paper but for scattering aerosol in the midlatitude simulation. Although there are minor differences in the strength of the gradient in heat fluxes, the resulting aerosol breeze, and the subsequent cloud formation, the overall impacts are qualitatively similar.

Supplementary Figure 1. Mean cross section through the domain of the scattering aerosol-midlatitude gradient simulation over all 12 hours of the simulation, averaged temporally and zonally. The abscissa is given as a function of distance from the domain center/maximum aerosol concentration (horizontal black line in Supplementary Figure 1b,c), such that the high-aerosol region is on the left and the low-aerosol region is on the right of these panels. Shading in (a) shows cloud condensate mixing ratios (g kg^{-1}), and in (b) the

downwelling shortwave flux (W m^{-2}). The wind barbs in (a) and (b) show the mean vertical and horizontal winds oriented along the aerosol gradient. The aerosol optical depth at the surface is represented in (c), while (d) depicts the sensible surface heat flux in red (left y-axis; W m^{-2}) and latent surface heat flux in blue (right y-axis; W m^{-2}), all averaged temporally and zonally.

We added a sentence describing these midlatitude tests in the Methods section, lines 287-293: “We also conducted identical simulations using a mean sounding over the central USA during July 2021 (corresponding to Case Study 1 in **Figure 1a-b**), with results shown in **Supplementary Figure 2-3**. The impact of the aerosol gradient in the midlatitude simulations is similar to that in the tropical simulations, with an aerosol breeze developing within a few hours. We have thus opted to present only the tropical simulations in the main text for simplicity, though the impact of different meteorological conditions on the properties of the aerosol breeze is certainly a valuable area of future study.”

Over the Maritime Continent, the effect of aerosol breeze is probably masked by the effect of sea breeze. What further confuses me is that the potentially important effect of land-sea contrast is not considered in the simulations.

For this study, we simply demonstrated for the first time that spatial heterogeneity in aerosol emissions alone (in the absence of gradients in any other surface properties, e.g., land-sea contrasts) is capable of driving mesoscale circulations which influence cloud and precipitation formation, and we examined the processes that may drive these circulations. We agree with the reviewer that there will be circumstances in which the aerosol breeze may interact with other mesoscale features, and that such interactions are worthy of future study. That being said, such interactions can only be properly understood once the fundamental processes governing the aerosol breeze are elucidated. In our concluding paragraph (lines 263-267), we specifically cite interactions with sea breezes in our recommended future work: “Our findings emphasize the need for greater consideration of aerosol breezes in future work, particularly in investigating the sensitivity of aerosol breezes to meteorology, aerosol type, land surface, varied spatial distributions, and interactions with other circulations (such as sea breezes or buoyant firestorm plumes).” However, examining such interactions in detail do not form a necessary part of this foundational study, which is intended as a proof-of-concept of the aerosol breeze acting in isolation.

Third, the treatment of solar radiation and aerosol concentration is questionable. The authors set the solar radiation to be constant at the solar noon. This might unrealistically exaggerate the effect of aerosol breeze. Furthermore, as shown in Figure 4, the effect of aerosol breeze is sensitive to the aerosol concentration. It seems reasonable to anticipate that the effect of aerosol breeze is also sensitive to the solar radiation.

The reviewer raises a fair point about the treatment of solar radiation. To address this, we have run a test with a realistic representation of the diurnal cycle, in which the simulation is started at 5AM (approximately two hours before local sunrise) and run until 7PM, after local sunset, for a total simulated time of 14 hours. **Supplementary Figure 4** shows that within two hours of the sunrise in this simulation (9-10AM LT), the aerosol breeze develops in a similar manner to that in the simulations presented in the paper where the zenith angle is fixed to solar noon. The aerosol breeze is maintained throughout the daylight hours. Thus, it does not seem that the fixed

solar zenith angle exaggerates the effect of the aerosol breeze. We have chosen not to present these simulations in the paper since it simplifies our analysis without impacting the main properties of the aerosol breeze shown. We have however added sentences describing the above in the Methods section, lines 321-323: “However, we also conducted a simulation with the diurnal cycle represented and found that a qualitatively similar aerosol breeze circulation developed within two hours of sunrise (**Supplementary Figure 4**).”, and included the following figure in the Supplementary Information.

Supplementary Figure 4. As in Supplementary Figure 1, but averaged only between 9-10AM (approximately two hours after sunrise) for the scattering aerosol, tropical simulation.

In addition, the aerosol concentration of the “reduced” simulations is not clear.

We have clarified the aerosol concentrations in the Methods section, lines 313-314: “In these reduced-concentration runs, the initial aerosol number concentration at the surface ranged from 50 to 500 cm^{-3} .”

Fourth, the two observational studies focus on smoke aerosols, which usually have a high percentage of light-absorbing aerosols. However, the aerosol in the simulation is assumed to be scattering, only.

As suggested by both reviewers, we have repeated the analysis presented in the paper for scattering aerosol (ammonium sulfate) but with absorbing aerosol (absorbing carbon) instead. A discussion of the findings from these simulations is included as a new section “Sensitivity to aerosol type” in lines 220-241. In summary, a gradient in absorbing aerosol drives a similar gradient in surface fluxes, thermal circulation, and gradient in clouds/precipitation. However, there are additional impacts to atmospheric stability due to the re-emitted longwave radiation which increase stability aloft and cap the cloud development at ~5km.

References:

Sokolowsky et al., 2022, which is cited at line 276, is not found in the references.

We have added this reference.

Reviewer #2 (Remarks to the Author):

Review on the “Aerosol breezes drive cloud and precipitation increases” by Leung and van den Heever

This paper argues that effect of the horizontal gradient of aerosols which is known as the aerosol breeze effect has long been underappreciated, however it can be a significant factor to better understand the aerosol-cloud interaction. The paper first shows two satellite observed cases to build the case, then it conducts a numerical study with a high-resolution model to show the aerosol breeze effect. The paper is well organized and well stated. The methodology is clear and is scientifically sounding.

Only major issue I found is the role of the absorbing aerosols, which is overlooked in the consideration throughout the manuscript. The study only considers scattering aerosols for the aerosol breeze effect. The two observation cases are indeed the wildfires, which naturally contain quite significant amount of light absorbing aerosols, such as black carbon and organic carbon aerosols. The paper should clearly state 1) how significant the role of the light absorbing aerosol in the aerosol breeze effect, 2) why wildfire cases are qualifying to demonstrate the role of the aerosol breeze cases. I would imagine that urban conditions are dominated by scattering aerosol and the aerosol breeze over those regions can be detected from satellite observations. I would like to suggest major revision for the Nature Communications upon the response to my major comment.

We thank the reviewer for their insightful comments and suggestions, particularly on the role of absorbing aerosol, which have greatly helped us to improve the quality of this manuscript. We provide responses below in blue font.

As suggested by both reviewers, we have repeated the analysis presented in the paper for scattering aerosol (ammonium sulfate) but with absorbing aerosol (absorbing carbon) instead. A discussion of the findings from these simulations is included as a new section “Sensitivity to aerosol type” in lines 220-243. In summary, a gradient in absorbing aerosol drives a similar gradient in surface fluxes, thermal circulation, and gradient in clouds/precipitation. However, there are additional impacts to atmospheric stability due to the re-emitted longwave radiation which increase stability aloft and cap the cloud development at ~5km. We have thus now demonstrated that the wildfire cases do represent the same aerosol breeze phenomenon by including results from model runs with absorbing aerosol, though we also note in lines 241-243 that “the specific characteristics of the aerosol breeze will likely depend on the dominance of scattering or absorbing aerosol in the region of interest.”

Minor comments

Line 52: Lee et al. (2014), citation is missing.
We have added this reference.

Line 80, 86-87 (Figure 1a,b,c): “low-aerosol” It needs to be more elaborated. Low-aerosol region is also cloud covered; therefore it is more-like missing AOD. I believe MODIS cannot observe

AOD in cloudy conditions. How one can tell AOD under cloudy conditions? Also I would like to suggest including wind vector to each case 1) to understand background conditions, and 2) to make sure if they (i.e., aerosol and cloud fields) are one dynamic system or two different systems.

The reviewer is correct that the cloud cover interferes with the MODIS AOD retrieval in the first observational case. This poses difficulty in identifying the impacts of aerosol gradients on clouds from satellite observations (and indeed in most satellite-based analyses of aerosol-cloud interactions). There is a clear-sky patch to the west of the shallow cloud field which shows the NE-SW gradient in aerosol from high to low AOD—we assume that the direction of the gradient is roughly similar across the image based on the wind field, since the prevailing wind is blowing uniformly from the high smoke region in the north. Thank you for your suggestion to include the wind barbs. We have now included these based on MERRA-2 for the same time period in the new Figure 1. These winds are generally in agreement with surface wind vectors taken from MesoWest (shown below). Winds are generally light throughout the whole domain. Smoke is also clear in the northern half of the visible imagery (Figure 1a), but dissipates moving south towards 38°N, approximately the same latitude as the shallow cloud field.

Surface wind vectors for Case 1 in Figure 1 (main text).

We have not added wind vectors to Case 1 (Figure 1c) since the domain is small relative to the size of reanalysis resolution.

Figure 2: Please show the unit of the wind barbs. Vertical wind looks quite weak.

We have added the unit of the wind barbs. As the reviewer notes, the mean vertical wind in the cross section is relatively weak ($\sim 0.05 \text{ m s}^{-1}$ in Figure 2). We note, however, that these are *mean* winds over 12 hours and over the whole domain. Individual updrafts have maximum vertical velocities of $3\text{-}5 \text{ m s}^{-1}$.

Figure 3-4: How significant the overall impact of aerosol breeze, e.g., compared to the impact of absorbing aerosols? Cloud fraction seems a useful variable to examine the aerosol breeze, however it is not considered in the result. I would suggest looking into cloud fraction in the revision.

As suggested by the reviewer, we have included cloud fraction as a metric in Figure 3d and Figure 4c.

Line 219-223: Again, light absorbing aerosols are missing in the discussion.

We have included a new section including absorbing aerosols. Please see our full response under the major comments above.

REVIEWER COMMENTS

Reviewer #1 (Remarks to the Author):

The revised manuscript is now substantially improved. I believe it is acceptable for publication.

Reviewer #2 (Remarks to the Author):

Review on the "Aerosol breezes drive cloud and precipitation increases" by Leung and van den Heever

The revised paper is significantly improved from the previous version, and I would recommend accept with minor revision. I have a few follow-up questions as in below.

Line 141-142: However, it seems me that the highest AOD is the left side of the figure, not center.

Line 143-144: Am I missing something? It seems me the 30W/m² lower shortwave is on the left side of the figure (0~10km of x-axis), not center.

Figure 2 and Line 155: It is thermally driven circulation however it never shows temperature gradient in Figure 2 and other places. Could you discuss more about the temperature gradient?

Line 303-308: This paper used LES model to test their aerosol breeze hypothesis. I wonder if regional or mesoscale models are already accounting for the Aerosol Breeze effect, or they are missing it as the paper is arguing? For example, if WRF-chem model runs in high horizontal resolution of 1km with similar aerosol configuration, then would it successfully repeat the LES simulation result or would it fail? If WRF-chem fails, then this study has found a very exciting new phenomenon. If WRF is also successful then the aerosol breeze is already incorporated in the existing model physics, but not identified yet. Right?

Reviewer #2 (Remarks to the Author):

Review on the “Aerosol breezes drive cloud and precipitation increases” by Leung and van den Heever

The revised paper is significantly improved from the previous version, and I would recommend accept with minor revision. I have a few follow-up questions as in below.

We thank the reviewer for their kind comments. Our responses and clarifications are below in blue font.

Line 141-142: However, it seems me that the highest AOD is the left side of the figure, not center.

The reviewer is correct that the highest AOD is on the left side of Figure 2. However, this represents the center of the model domain (Figure M1b). As we describe in the figure caption, lines 129-133: “The abscissa is given as a function of distance from the domain center/maximum aerosol concentration (horizontal black line in Figure M1b,c), such that the high-aerosol region is on the left and the low-aerosol region is on the right of these panels.”

However, this said, we recognize that this may be confusing to readers, so we have added a sentence clarifying this in the text (lines 141-145): “Note that while the high-aerosol region is in the center of the model domain (**Figure M1b**), the cross-section in **Figure 2** is constructed such that the abscissa is given in terms of distance from the center line where aerosol concentrations are highest (i.e., the domain has been “folded” across the black center line indicated in **Figure M1b** to more clearly demonstrate the circulation).”

Line 143-144: Am I missing something? It seems me the 30W/m² lower shortwave is on the left side of the figure (0~10km of x-axis), not center.

Please see the response to the previous question. To avoid confusion, we have also now changed lines 134-136 to refer to the high and low aerosol regions, rather than the center/edges of the domain: “As a result of the increased light extinction, the downwelling shortwave at the surface is ~30 W m² lower in the high-aerosol region compared to the low-aerosol region (**Figure 2b**).”

Figure 2 and Line 155: It is thermally driven circulation however it never shows temperature gradient in Figure 2 and other places. Could you discuss more about the temperature gradient? This is a good point. Thank you for this suggestion. We have now included the surface potential temperature in Figure 2 and 5, as well as in the figures in the supplement. Lines 170-174 now also read: “As a result of the increased light extinction, the downwelling shortwave at the surface is ~30 W m² lower in the high-aerosol region compared to the low-aerosol region (**Figure 2b**). This causes uneven heating of the land surface, with greater surface temperatures occurring in the low-aerosol region (**Figure 2c**). A gradient in surface fluxes forms opposite to the direction of the gradient in aerosol (**Figure 2d**).”

Line 303-308: This paper used LES model to test their aerosol breeze hypothesis. I wonder if regional or mesoscale models are already accounting for the Aerosol Breeze effect, or they are missing it as the paper is arguing? For example, if WRF-chem model runs in high horizontal

resolution of 1km with similar aerosol configuration, then would it successfully repeat the LES simulation result or would it fail? If WRF-chem fails, then this study has found a very exciting new phenomenon. If WRF is also successful then the aerosol breeze is already incorporated in the existing model physics, but not identified yet. Right?

It is certainly conceivable, and should in fact be expected, that any model with appropriate representations of cloud, radiation, land surface and aerosol processes and with grid spacings on the order of 1km should reproduce these aerosol breezes (as the reviewer suggests) *as long as* they are properly representing the gradient in aerosols rather than assuming homogenous aerosol concentrations. Unfortunately, as we outline in the introduction, almost all simulations reported in the literature assume the latter, from high-resolution idealized simulations through NWP and GCMs. As such, these simulations will not capture aerosol breezes.

We have edited lines 197-201 to clarify this: “Furthermore, this result suggests that unrepresented aerosol heterogeneities across climate and regional models with grid spacings that are coarse relative to the scale of the aerosol gradient may lead to biases in rain timing, distribution, and even total rain amount. Limited area models at higher resolutions may be able to capture the effect of the mesoscale secondary circulations described here, but only if the spatial gradients of aerosol concentration are properly represented.”

Similarly, lines 275-279 now read: “It may also lead to biases in the estimation of aerosol-cloud-precipitation feedbacks in other higher-resolution studies which do not appropriately represent the spatial heterogeneity of aerosol emissions, either because of sub-grid scale gradients or because the aerosol concentration is otherwise assumed to be homogenous across the domain.”